# A statistical study of the $O_2Atm(0-0)$ aurora observed by the Swedish satellite MATS

Judit Pérez-Coll Jiménez<sup>1</sup>, Nickolay Ivchenko<sup>1</sup>, Ceona Lindstein<sup>1</sup>, Lukas Krasauskas<sup>2</sup>, Jonas Hedin<sup>2</sup>, Donal P. Murtagh<sup>3</sup>, Linda Megner<sup>2</sup>, Björn Linder<sup>2</sup>, and Jörg Gumbel<sup>2</sup>

**Correspondence:** Judit Pérez-Coll Jiménez (juditpcj@kth.se)

#### Abstract.

This study conducts a statistical analysis of the aurora observed by the Swedish satellite MATS. MATS' main instrument is a telescope that performs limb imaging at six different wavelength intervals, among them the 762 nm wavelength emission in the  $O_2$  atmospheric band. This emission, even though it can not be observed from the ground, is important at mesosphere/lower thermosphere altitudes for both atmospheric airglow and aurora. Here, some auroral properties of this emission, such as peak altitude, geomagnetic location, and auroral intensity, are examined and compared to the SME and Kp geomagnetic indices. A total of 378 events are analyzed. An average geomagnetic latitude of 67.7° is found in both hemispheres, and an average peak altitude of 103 km is obtained. The peak altitude shows dependence on the magnetic local time. Auroral intensities of the order of  $10^2$ - $10^3$  kR are observed.

#### 0 1 Introduction

The  $O_2$  atmospheric band emission at 762 nm or  $O_2$ Atm(0-0) is produced by radiative transition of the  $O_2(b^1\Sigma_g^+, v=0)$  state down to the  $O_2(X^3\Sigma_g^-, v=0)$  fundamental state. This emission is known to be one of the main emissions in atmospheric airglow (Chamberlain, 1995) and one of the least understood auroral emissions, as we discuss below.

The main mechanism that produces the  $O_2(b^1\Sigma_g^+,v=0)$  state during aurora is based on the energy transfer from the  $O(^1D)$  state to the  $O_2(X^3\Sigma_g^-,v=0)$  ground state (Deans et al., 1976; Wallace and Chamberlain, 1959), while excitation of the ground state by direct electron impact has been found to contribute less than 5% to the total  $O_2Atm(0-0)$  emission (Feldman, 1978).  $O(^1D)$  is produced in aurora mainly by the impact of secondary electrons on atomic oxygen and dissociative recombination of  $O_2^+$  (Solomon et al., 1988). At mesopause altitudes, where the airglow originates, the main sources of  $O(^1D)$  during daytime are the dissociation of ozone and molecular oxygen (Grygalashvyly et al., 2021). During nighttime, several weak of sources of  $O(^1D)$  had been identified, but the Kalogerakis-Sharma mechanism (Sharma et al., 2015; Kalogerakis et al., 2016) has recently been proposed to be the major source of  $O_2(b^1\Sigma_g^+, v=0)$  (Kalogerakis, 2019). According to this mechanism, nighttime  $O(^1D)$  is populated from the five highest vibrationally excited hydroxyl levels.

<sup>&</sup>lt;sup>1</sup>Division of Space and Plaea Physics, KTH-Royal Institute of Technology, Stockholm, Sweden

<sup>&</sup>lt;sup>2</sup>Department of Meteorology (MISU), Stockholm University, Stockholm, Sweden

<sup>&</sup>lt;sup>3</sup>Earth and Space Sciences, Chalmers University of Technology, Göteborg, Sweden

Direct observations of the  $O_2Atm(0-0)$  band are not possible from the ground due to the absorption by  $O_2$  below the emitting region. Some measurements have been inferred from other  $O_2Atm$  bands of higher vibrational levels (Vallance Jones and Gattinger, 1973; Gattinger and Jones, 1974). However, the vibrational development of the system is still uncertain, and the inferred values are inconsistent with measurements in situ (Deans et al., 1976; Gattinger et al., 2010).

Rocket observations of the auroral  $O_2Atm(0-0)$  emission have been studied by Deans et al. (1976), Feldman (1978) and McDade et al. (1985). Deans et al. (1976) measured a height integrated intensity of the 762 nm emission above 90 km of 28 kR, with an observed ratio with the  $N_2^+$  at 427.8 nm of I(762)/I(427.8) = 4.5, and the peak volumetric emission rate of the  $O_2Atm(0-0)$  band emission was found to be around 105 km altitude. Feldman (1978) found similar values for the integrated 762 nm emission and its peak altitude, but the value of the ratio I(762)/I(427.8) was higher and increased monotonically with decreasing altitude below 120 km instead of decreasing, as found in Deans et al. (1976). McDade et al. (1985) reported measurements of the  $O_2Atm(0-0)$  emission during pulsating aurora. They reported an auroral component of 2 kR, and a ratio I(762)/I(427.8) of  $\sim 0.3$ , revealing the high variability of the  $O_2Atm(0-0)$  emission intensity.

Even though  $O_2$ Atm(0-0) aurora must have also been observed from most satellites able to measure airglow, surprisingly little has been published on the topic. Gattinger et al. (2010) published a study on auroral emissions measured by the OSIRIS spectrograph on board the Odin spacecraft. All wavelengths from 275 to 815 nm were exposed simultaneously, and the auroral spectrum averaged over limb tangent altitudes from 100 to 105 km was obtained. This study revealed the  $O_2$ Atm(0-0) emission at 762 nm to be the brightest feature in the spectrum when viewed from space, measuring a limb brightness of 4000 kR and a ratio I(762)/I(427.8) = 19.4.

Here, a statistical study of  $O_2Atm(0-0)$  aurora using measurements obtained from the limb instrument on the MATS satellite is presented. In Section 2, the main instrumentation used is described, followed by an explanation of the analysis performed on the measured data in Section 3. In the results, the Magnetic Local Time (MLT) / magnetic latitude distribution of all auroral events is presented, together with the associated SME index. A statistical analysis of the peak altitude in relation to the MLT and the auroral intensity is also performed. Finally, the discussion is presented in Section 5, and the conclusions are stated at the end.

## 2 Instrumentation

35

The primary goal of the Swedish MATS (Mesospheric Airglow/Aerosol Tomography and Spectroscopy) satellite mission is to determine the global distribution of gravity waves and other structures in the mesosphere and lower thermosphere region. MATS was launched on November 4, 2022, into a sun-synchronous orbit at an altitude of 585 km. It has nominal equator passages at around 05:30 and 17:30 local time, and its main targets are airglow in the  $O_2$  atmospheric band and sunlight scattered from noctilucent clouds (Gumbel et al., 2020). The main instrument on the MATS satellite is the limb instrument, which provides a horizontal (limb) view of the atmosphere between 75 and 115 km altitude.

The Limb instrument uses a single off-axis three-mirror reflective telescope with a focal ratio of f/7.3 and an entrance pupil diameter of 35 mm to capture a 260 km  $\times$  40 km atmospheric scene at the tangent point (Megner et al., 2025; Hammar et al.,

2019). It has six spectral channels, two in the ultraviolet with central wavelengths at 270 and 304.5 nm for the measurement of NLC and four in the near-infrared with central wavelengths at 762, 763, 754, and 772 nm for the measurement of airglow and aurora (Gumbel et al., 2020). The 763 nm channel or "IR2", which we use in this study, has a bandwidth of 8 nm and a sampling of approximately  $5.7 \text{ km} \times 290 \text{ m}$  (horizontally/vertically) at the tangent point. The IR2 filter transmission wavelengths include almost the entire emission band (98.1% at 200 K rotational temperature, 96.9% at 250 K, and 95.4% at 300 K).

MATS takes an image every 6 seconds. The calibration of the images uses a parameterized model that takes into account transmissivity of the optics, dark current, and quantum efficiency variations so that the image pixel values display the actual measured radiance (Gumbel et al., 2020; Megner et al., 2025).

## 3 Analysis

The dataset used in this study spans from February 8 to April 30, 2023, corresponding to the period when the satellite was in continuous operation with nominal pointing. Data with tangent point latitude above 45 ° in both hemispheres are used in the study.

Figures 1a-d show an example of four limb images taken by the limb instrument. The spectral intensity has been converted to auroral intensity (in Rayleighs) by multiplying it by  $4\pi 10^{-10}$  and the bandwidth of the filter in nm (8 nm for IR2). In these images, the Earth is located below the field of view. In a) and d), the airglow layer can be observed around pixel row 120, while aurora can be seen in b) and d) as irregular increases in brightness. To visualize the time sequence of the observations, the central column of each consecutive image is put next to the following one to form a keogram (see panel e). Auroral structures can be distinguished in the keogram as parabolic-shaped increases in radiance imposed over the constantly present airglow layer. This is explained by the perspective of the aurora in relation to the orbit of MATS: aurora appears from the bottom of the image as the satellite approaches it (see Figure 1g)i.), it reaches a peak when viewed from the tangent point (ii.), and it disappears again to the bottom of the image as the auroral structure gets out of the instrument's field of view (iii.).

While the airglow layer is always present in the images, its brightness and position change in the course of the orbit. To remove the airglow contribution from the keograms in order to detect the emissions related to the aurora, a third-order polynomial regression is applied to all the keogram rows above 130. The regression fits the background, excluding the auroral data points. This fit is then subtracted from the entire row of data. In Figure 1f the pixel values of three of the keogram rows in panel e) have been plotted, and a solid line of the same color indicates the background subtracted from the data. In this particular case, the background variation is small. However, when the satellite moves into the dayside or nightside, the gradient is significant, and a third-order polynomial is needed for the fit.

An automated algorithm was developed to detect the aurora. The algorithm searches for peaks in the intensity altitude profile above the airglow layer. The top of the airglow and, thus, the lower limit for auroral detection is taken at row 150, corresponding to an altitude of about 95 km. Keograms for each orbit and hemisphere were generated. For each column of each keogram, the algorithm verifies that its average brightness (over all the pixels above row 150) is above  $15 \cdot 10^{13}$  ph· nm<sup>-1</sup>· m<sup>-2</sup>· sr<sup>-1</sup>· s<sup>-1</sup>. To ensure that the brightness above the threshold is not due to the upper edge of the airglow layer, the algorithm also verifies that

**Figure 1.** a)-d) Processed images taken by the limb instrument at the times indicated on top of each panel. The colormap limits are the same as in panel e). e) Keogram created by adding the central column of each image in chronological order. X-axes indicate time and latitude of the tangent point (TPLat). Y-axes represent the row number and the average (among all times) altitude of the tangent point corresponding to each pixel row (TPAlt). Vertical red lines indicate the columns corresponding to the images in panels a)-d). Horizontal lines correspond to the rows plotted in the next panel. f) Pixel values of three rows of the keogram (dots) together with the fitted background (solid line). g) 2D sketch of MATS perspective of aurora (not to scale), corresponding to the peak of the parabola in the keogram (ii.), and when observed below the airglow before the peak (i.), and after (iii.). The grey area represents the instrument's field of view, while the white line is the tangent line.

even 10 rows above the top of the airglow, the brightness is above  $18 \cdot 10^{13}$  ph· nm<sup>-1</sup>· m<sup>-2</sup>· sr<sup>-1</sup>· s<sup>-1</sup>. If these two conditions are met, the brightest pixel of the column is selected, and it is considered a candidate sighting of aurora. The threshold values were decided after carefully inspecting several keograms with and without aurora to achieve a balance between the detection sensitivity and the frequency of spurious detection due to noise. Note that the detection algorithm is run on images taken with the filter IR1.

Potential auroral sightings have several sequential images with an additional brightness peak above the airglow. The time when the altitude of the peak is at its highest corresponds to viewing the peak of the auroral intensity at the tangent point. In the algorithm, the pixels selected as auroral candidates are grouped as long as they are less than 4 minutes apart, and groups with fewer than four pixels are discarded from the database. Then the time of the pixel with the highest row number within a group is taken as a representative of that auroral event. These times are verified visually, discarding the events where the keogram appearance is unclear. The altitude, latitude, and time of the auroral event are taken from the peak brightness pixel. For each auroral event, the brightness is taken as an average of a 3x5 pixel area surrounding the peak brightness pixel. This is done in the images taken with IR2, with the background subtracted as outlined above. It should be noted that the parameters are derived from a single image when the aurora is seen at the tangent point, i.e., there are no assumptions about the aurora not varying in time.

#### 4 Results

100

115

120

A total of 378 events were analyzed, 146 of which occurred in the northern hemisphere (NH) and 232 in the southern hemisphere (SH). Figure 2 shows the MLT/magnetic latitude disposition and the SuperMAG Electrojet (SME) index associated with each of the events (see description of the SME index by Newell and Gjerloev (2011a, b)). The black paths indicate the geomagnetic location of the limb instrument's tangent point. With a local time of ascending node of 17:30, the orbit passes on the dayside of the pole in the northern hemisphere. As seen in Figure 2, the satellite has very limited measurements in the night portion of the auroral oval (19:00 to 5:00 MLT) in the northern hemisphere, compared to the southern hemisphere. The wider spread in the southern hemisphere is because the magnetic south pole is located at a lower latitude than the magnetic north pole.

The events are mostly observed between  $60^{\circ}$  and  $75^{\circ}$  magnetic latitude, with an average at  $\pm$  67.5°. Regarding the distribution in relation to their SME index, Figure 2 shows that, in both hemispheres, MATS observed auroral events with a higher SME index at lower latitudes, while events with a lower SME index were more common at higher latitudes. The slope value of the fitted line in panels c and d indicates a steeper relation in the northern hemisphere.

Figure 3a shows the altitude distribution of the events as a function of MLT. The data are binned according to hemisphere (SH/NH), Kp-index (low/moderate/high), and MLT (one-hour bins). The average values have been plotted together with the standard error of the mean for each bin with more than two events. Horizontal lines indicate the average altitude for all events in each Kp-index bin. The total average of all events analyzed is around 103 km altitude. The average altitude for events with a Kp-index between 6 and 9 is slightly higher than for lower Kp. As mentioned before, because of the satellite's orbit, only

Figure 2. a)-b) Distribution of the analyzed events in MLT plots for the northern hemisphere (a) and southern hemisphere (b). Color indicates the SME index associated with each event. c)-d) Scatter plot of the Magnetic Latitude vs the SME index for the northern hemisphere (c) and southern hemisphere (d). The solid represents a fit of the data. The value of it's slope is indicated in the legend.

events in the southern hemisphere can be seen during the night hours, while more northern hemisphere data points are visible in the morning and evening hours. A notable feature of this plot is the lower altitude of events between 5 and 7 MLT in the southern hemisphere.

Figure 3b shows a scatterplot of the peak intensity vs the peak altitude for all events analyzed, with different colors and markers for different bins of Kp-index and hemispheres, respectively. Most events have altitudes between 95 and 110 km, and the intensities range between 200 and 5500 kR.

**Figure 3.** a) Average altitude of events occurring within the same one-hour bin as a function of MLT. Asterisks represent the northern hemisphere, while triangles represent the southern hemisphere. Blue includes events with Kp-index between 0 and 3, green between 3+ and 6, and red between 6+ and 9. Error bars indicate the standard error of the mean. Horizontal dashed lines indicate the averaged altitude of all events in each of the Kp-index bins. The mean values are indicated in the legend together with the uncertainty of the mean. b) Intensity as a function of altitude for all events analyzed. Colors and markers follow the same pattern as in a).

## 5 Discussion

130

135

This work showed a clear dependence between latitude and SME index (see Figure 2) of 762 nm aurora. Events associated with a weaker activity of the auroral electrojet (lower SME index) were found to be closer to the poles, while events associated with a stronger activity of the auroral electrojet (higher SME index) were located at lower latitudes. This agrees with the results of the study on  $O_2$  aurora at the 1.27  $\mu$ m emission by Gao et al. (2020), who found that the  $O_2$  auroral zone expands equatorward with the enhancement of the geomagnetic activity. The result also follows the well-known expansion of auroral ovals to lower latitudes during high geomagnetic activity (Kivelson and Russel, 1995). The dependence found in this study was -5.5· $10^{-3}$  °/nT in the Northern Hemisphere and  $4.5\cdot10^{-3}$  °/nT in the Southern hemisphere.

Auroral intensities or limb brightness measured in this study are within the range of 200 to 5500 kR, with the majority of the events centered around 500 kR. The obtained values are generally lower than, but comparable to, the limb brightness measured by OSIRIS as reported in Gattinger et al. (2010).

The altitude of the events was analyzed for different magnetic local times, hemispheres, and Kp-indices in Figure 3a. The average altitude was found at about 103 km altitude, which is close to the values reported in the rocket experiments by Deans et al. (1976) and Feldman (1978). All three Kp bins show very close values of the average altitude of the peak. The small differences - if significant - may be related to the combination of the auroral distribution and orbital coverage, rather than to global changes in the auroral characteristics as such. For magnetic local times of 05:00 to 07:00, the average altitude of the peak in the southern hemisphere tends to be below average. This may be related to the precipitation of energetic electrons causing pulsating aurora, which have been found to be more prominent in the morning sector, and would produce lower peak altitudes (Partamies et al., 2022).

In Figure 3b, the spectral intensity was plotted as a function of altitude. While events with low auroral intensities seem to have peak altitudes evenly spread between 95 and 110 km, events with high auroral intensities seem to decrease with increasing altitudes. The Knight relation (Knight, 1973) demonstrates that field-aligned electron beams with higher energy flux are associated with higher characteristic electron energies. This can explain our results, since higher electron flux results in brighter aurora, and higher electron energy will lead to electrons reaching lower altitudes. Note that in this study, the latitudinal extent of the aurora is not resolved from the images, which can affect the results. Gao et al. (2020) performed a long-term study on  $O_2$  aurora based on observations of the emission at  $1.27\mu$ m. It was found that, for Kp Levels from 1 to 5, the auroral peak height varies between 104 and 112 km. In our study, peak heights have been found down to about 95 km. This discrepancy might be related to the different ways of handling the removal of the airglow layer.

It is important to mention some limitations concerning this work. Our detection algorithm worked well when finding clear and isolated events, but may have missed some events with a more complex spatial distribution of the aurora, e.g., with multiple adjacent large-scale structures. During the manual visualization of the detected events, some unclear cases were removed from the database. In some clear cases of aurora, the highest pixel had been automatically selected close to the edges instead of the actual auroral peak and had to be corrected manually. An estimation indicates that in about 90% of the auroral oval crossings, the aurora is not visible or too weak to be detected by the algorithm. Then, about 2% of the crossings detect complex auroral distributions (indicating closely spaced multiple auroral structures) that are omitted, while about 8% show clearly detectable aurora. It is also relevant to mention that due to operational reasons, there are gaps in the dataset of up to 15%. Some bias might also have been introduced when removing the background airglow from the auroral events happening during specific times when the satellite moved into or out of the sunlit portion of the orbit. For limb observations alone, it is difficult to estimate the auroral "thickness", i.e., the horizontal extent along the orbit. This makes it difficult to quantitatively interpret the observed intensity values or compare them to vertical profiles reported from the rocket experiments. A tomographic reconstruction of the aurora could retrieve a 3D distribution of the auroral intensity. This goes beyond the scope of the current study. Besides, this would require the aurora not to vary during the time it is observed by the instrument, about three minutes, which is often not the case. This study relies on the "snapshots" of the aurora when it is at the tangent point in the field of view.

We used the IR2 channel data in this work. This filter is centered at 763 nm and captures the whole (0,0) band. Its bandwidth is 8 nm, which can lead to some pollution from other wavelengths. In future analysis, combining all four IR limb imaging channels can be used to increase the accuracy of the measurements.

## 6 Conclusions

150

160

165

A statistical study of some properties of O<sub>2</sub>Atm(0-0) aurora as measured by the limb instrument on board the MATS satellite has been presented. The auroral events were distributed between 60 and 75 degrees latitude, with events with higher Kp-index located at lower latitudes and events with lower Kp-index closer to the poles, as expected. The average altitude of all the events was 103 km, and lower altitudes were observed for aurora in the post-midnight sector. Limb auroral brightness values between 200 and 5000 kR were observed.

180 Data availability. The MATS level 1b dataset (as used in this study) is stored in the Bolin centre database https://bolin.su.se/data/mats-level-1b-limb-cropd-1.0.

Author contributions. The manuscript was primarily written by JPCJ and NI with contributions and comments from the other coauthors. The auroral detection algorithm was developed by CL. The analysis, visualization and cleaning of the data, as well as the plots and sketches were done by JPCJ. The MATS science mission was conceptually developed, and its instruments were designed by JG, LM, DPM, NI and JH, among others.

Competing interests. No competing interests are present.

185

Acknowledgements. This work has been financed by the Swedish National Space Agency under the grants 2020-00154, 22/15, 298/17 (Royal Institute of Technology), 21/15, 297/17, 2021-04876, 2022- 00108 (Stockholm University), and 23/15, 299/17 (Chalmers University of Technology). The MATS satellite project has been financed by the Swedish National Space Agency under grant 2021-00052.

210

215

- Chamberlain, J. W.: The Airglow Spectrum, chap. 9, pp. 345–392, American Geophysical Union (AGU), https://doi.org/10.1002/9781118668047.ch9, 1995.
- Deans, A. J., Shepherd, G. G., and Evans, W. F. J.: A rocket measurement of the  $O_2(b^1\Sigma + -X^3\Sigma -)(0-0)$  Atmospheric Band in Aurora, Journal of Geophysical Research (1896-1977), 81, 6227–6232, https://doi.org/10.1029/JA081i034p06227, 1976.
- Feldman, P. D.: Auroral excitation of optical emissions of atomic and molecular oxygen, Journal of Geophysical Research: Space Physics, 83, 2511–2516, https://doi.org/10.1029/JA083iA06p02511, 1978.
  - Gao, H., Xu, J., Chen, G.-M., Zhu, Y., Liu, W., and Wang, C.: Statistical Structure of Nighttime O2 Aurora From SABER and Its Dependence on Geomagnetic and Solar Activities in Winter, Journal of Geophysical Research: Space Physics, 125, e2020JA028302, https://doi.org/https://doi.org/10.1029/2020JA028302, 2020.
- Gattinger, R. L. and Jones, A. V.: Quantitative Spectroscopy of the Aurora. II. The Spectrum of Medium Intensity Aurora Between 4500 and 8900Å, Canadian Journal of Physics, 52, 2343–2356, https://doi.org/10.1139/p74-305, 1974.
  - Gattinger, R. L., Vallance Jones, A., Degenstein, D. A., and Llewellyn, E. J.: Quantitative spectroscopy of the aurora. VI. The auroral spectrum from 275 to 815 nm observed by the OSIRIS spectrograph on board the Odin spacecraft, Canadian Journal of Physics, 88, 559–567, https://doi.org/10.1139/P10-037, 2010.
- Grygalashvyly, M., Strelnikov, B., Eberhart, M., Hedin, J., Khaplanov, M., Gumbel, J., Rapp, M., Lübken, F.-J., Löhle, S., and Fasoulas, S.: Nighttime O(<sup>1</sup>D) and corresponding Atmospheric Band emission (762 nm) derived from rocket-borne experiment, Journal of Atmospheric and Solar-Terrestrial Physics, 213, 105 522, https://doi.org/10.1016/j.jastp.2020.105522, 2021.
  - Gumbel, J., Megner, L., Christensen, O. M., Ivchenko, N., Murtagh, D. P., Chang, S., Dillner, J., Ekebrand, T., Giono, G., Hammar, A., Hedin, J., Karlsson, B., Krus, M., Li, A., McCallion, S., Olentšenko, G., Pak, S., Park, W., Rouse, J., Stegman, J., and Witt, G.: The MATS satellite mission gravity wave studies by Mesospheric Airglow/Aerosol Tomography and Spectroscopy, Atmospheric Chemistry and Physics, 20, 431–455, https://doi.org/10.5194/acp-20-431-2020, 2020.
  - Hammar, A., Park, W., Chang, S., Pak, S., Emrich, A., and Stake, J.: Wide-field off-axis telescope for the Mesospheric Airglow/Aerosol Tomography Spectroscopy satellite, Appl. Opt., 58, 1393–1399, https://doi.org/10.1364/AO.58.001393, 2019.
  - Kalogerakis, K. S.: A previously unrecognized source of the O<sub>2</sub> Atmospheric band emission in Earth's nightglow, Science Advances, 5, eaau9255, https://doi.org/10.1126/sciadv.aau9255, 2019.
    - Kalogerakis, K. S., Matsiev, D., Sharma, R. D., and Wintersteiner, P. P.: Resolving the mesospheric nighttime 4.3  $\mu$ m emission puzzle: Laboratory demonstration of new mechanism for OH(v) relaxation, Geophysical Research Letters, 43, 8835–8843, https://doi.org/10.1002/2016GL069645, 2016.
- Kivelson, M. G. and Russel, C. T.: Aurora and auroral ionosphere, pp. 477–481, Cambridge University Press, University of California, Los Angeles, https://doi.org/10.1017/9781139878296, 1995.
  - Knight, S.: Parallel Electric Fields, Planetary and Space Science, 21, 741–750, https://doi.org/10.1016/0032-0633(73)90093-7, 1973.
  - McDade, I. C., Llewellyn, E. J., and Harris, F. R.: A rocket measurement of the (0–0) atmospheric band in a pulsating aurora, Canadian Journal of Physics, 63, 1322–1329, https://doi.org/10.1139/p85-218, 1985.
- Megner, L., Gumbel, J., Christensen, O., Linder, B., Murtagh, D., Ivchenko, N., Krasauskas, L., Hedin, J., Dillner, J., Giono, G., Olentsenko, G., Kern, L., and Stegman, J.: The MATS satellite: Limb image data processing and calibration, https://doi.org/10.5194/egusphere-2025-265, 2025.

- Newell, P. T. and Gjerloev, J. W.: Substorm and magnetosphere characteristic scales inferred from the SuperMAG auroral electrojet indices, Journal of Geophysical Research: Space Physics, 116, https://doi.org/https://doi.org/10.1029/2011JA016936, 2011a.
- Newell, P. T. and Gjerloev, J. W.: Evaluation of SuperMAG auroral electrojet indices as indicators of substorms and auroral power, Journal of Geophysical Research: Space Physics, 116, https://doi.org/10.1029/2011JA016779, 2011b.
  - Partamies, N., Whiter, D., Kauristie, K., and Massetti, S.: Magnetic local time (MLT) dependence of auroral peak emission height and morphology, Annales Geophysicae, 40, 605–618, https://doi.org/10.5194/angeo-40-605-2022, 2022.
  - Sharma, R. D., Wintersteiner, P. P., and Kalogerakis, K. S.: A new mechanism for OH vibrational relaxation leading to enhanced CO<sub>2</sub> emissions in the nocturnal mesosphere, Geophysical Research Letters, 42, 4639 4647, https://doi.org/10.1002/2015GL063724, 2015.
- Solomon, S. C., Hays, P. B., and Abreu, V. J.: The auroral 6300 Å emission: Observations and modeling, Journal of Geophysical Research: Space Physics, 93, 9867–9882, https://doi.org/10.1029/JA093iA09p09867, 1988.
  - Vallance Jones, A. and Gattinger, R. L.: Indirect Excitation Processes in Aurora, in: Physics and Chemistry of Upper Atmosphere, edited by McCormac, B. M., pp. 232–240, Springer Netherlands, Dordrecht, ISBN 978-94-010-2542-3, 1973.
- Wallace, L. and Chamberlain, J.: Excitation of O<sub>2</sub> atmospheric bands in the aurora, Planetary and Space Science, 2, 60–70, https://doi.org/10.1016/0032-0633(59)90060-1, 1959.