# Peer review of "A statistical study of the $O_2Atm(0-0)$ aurora observed by the Swedish satellite MATS"

_EGUsphere, 2025_

## Author Response (AR1)

**Major comment**

As stated above, currently the main results presented in the manuscript do not really bring much novel scientific insight in O2 auroral emissions. The fact that the auroral oval expands with increasing geomagnetic activity is very well-known and documented; the 762 nm aurora peak altitude and brightness are found to be similar to those derived in past studies. However, with the unique new dataset provided by the MATS mission, and given the scarcity of the literature on O2 auroral emissions, there is a lot of room for making new findings. I may suggest a few avenues that the Authors could consider to get there, some of which would require expanding slightly the statistical treatment of the data, while others could in fact mostly rely on surveying the existing literature.

Thank you for your valuable suggestions on how to increase the quality of our study. We have decided to implement some of your suggestions in this study, and will keep the rest in mind for future publications. We expect to develop a tomography algorithm in the future that will allow us to perform a deeper analysis of the data. We think of this publication as a first glance at the data, showcasing a new satellite that can be useful for this kind of study in a concise letter.

* Could the altitude thickness of the auroral emissions be evaluated (using e.g. the full width at half maximum as the metric) and be a parameter studied as a function of MLT, geomagnetic latitude, Kp index, spectral intensity, etc.?

Evaluating the vertical extent of the auroral emissions could be considered for future publications. However, there are a couple of concerns that make it challenging to do for this manuscript. First, to evaluate the full width at half maximum, we rely on a precise level of the background ("zero level"). The challenge of removing the background is the presence of the airglow and distinguishing it from the aurora. The paper presents how this is done in our case, and in combination with the threshold, this gives an objective way of selecting clear events of the aurora. The overlap of aurora and airglow below the peak affects where exactly the "half maximum" is located, and any uncertainty of the background (airglow) subtraction will contribute to the uncertainty of the altitude extent. In addition, the vertical extent of our images is limited and, even though the maximum altitude of emission is generally inside the field of view, in many cases the brightness of the emissions at the top of the image may still be above the half of the peak brightness, making it impossible to evaluate the vertical extent of the auroral layer in these cases.

* Could another geomagnetic index than Kp be considered – for instance an auroral electrojet such as AE or SME? These might be more insightful than Kp if looking at the properties of the aurora such as peak altitude and brightness, as they respond specifically to auroral/substorm activity.

Thank you for your suggestion. We agree that these geomagnetic indices are more insightful when studying the aurora. We have decided to substitute the Kp index with the SME index, as it is a better representative of the auroral current strength.

* Rather than a mere qualitative assessment that the O2 762 nm auroral emission geomagnetic latitude decreases with increasing Kp index, could a parametrisation of this latitude as a function of Kp (or other index) be derived? It could then be discussed with respect to the latitudinal dependence of other auroral emission lines on geomagnetic activity, enabling one for instance to determine whether O2 762 nm occurs in a specific part of the auroral oval and is associated with a certain type of particle precipitation.

Thank you for your suggestion. We have changed the plots in Figure 2 to show the SME index instead of the Kp index. We have also included two additional plots showing the MLat - SME index dependence for both hemispheres.

[Figure]

**Figure 2.** a)-b) Distribution of the analyzed events in MLT plots for the northern hemisphere (a) and southern hemisphere (b). Color indicates the  SME index associated with each event. c)-d) Scatter plot of the Magnetic Latitude vs the SME index for the northern hemisphere (c) and southern hemisphere (d). The solid represents a fit of the data. The value of it's slope is indicated in the legend.

We wanted to do a comparison with the OVATION-PRIME model, but it has resulted in being somewhat challenging, since the model only uses solar-wind data as input, and not geomagnetic indices as we had thought. For a statistical comparison the model would been to be run for all the events observed, which we feel is outside the scope of thi spaper.

* Can any interhemispheric asymmetries be noted when the events do not take place close to the equinox?

MATS's orbital parameters combined with the configuration of the magnetic field introduce a significant amount of biases to studying interhemispheric asymmetries, as the satellite does not have symmetrical passes in both hemispheres. For this reason, evaluating any interhemispheric features would require a separate study (and preferably a satellite in a non-sunsynchronous orbit - covering a range of local times), which is beyond the scope of this paper.

\* How do the results – in terms of O2 762 nm auroral emission altitude, MLT and geomagnetic latitude distribution, brightness – compare to what is known about other auroral emission lines?

While it would be good to have a full comparison of different parameters among several emissions, we believe that only studies conducted by limb imaging would be significant for our work (ideally co-located from the same platform). These, however, are very scarce. In addition, we are aiming to publish a short communication (letter), which limits how much we can add to the current manuscript. We have included a small comparison with Gao et al. as suggested by the reviewer, but have failed to find other publications relevant for direct comparison.

\* Would it be insightful to look into the cross-oval extent of the detected auroral events (again as a function of MLT, geomagnetic latitude, geomagnetic activity)? Could this be estimated by evaluating the width of the parabola associated with the signature of auroral emissions in keograms?

In the future, a three-dimensional tomographic reconstruction of the events is planned, which would provide a full approach to this problem and several others. This, however, relies on the validated tomographic reconstruction algorithm (which is currently work in progress applied to MATS data). The algorithm needs certain adaptations for auroral tomography compared to airglow tomography, as the aurora is not earth-fixed in the way the airglow is. Finally, it only works for cases where the emission distribution is sufficiently stable in time, an assumption which is often violated in the aurora. Thus, we consider this topic to be out of the scope of this paper, it is difficult to pursue on a statistical basis at this stage..

\* Is it possible to get additional information by analyzing the full images rather than keograms? In Fig. 1b–c, there seems to be some structuring of the auroral emissions visible along the horizontal direction of the images. Making use of the 2D nature of the MATS data could lead to findings that were not possible from rocket measurements, which are intrinsically 1D.

Yes, that is an excellent point. There is more information in the full images than what we can extract from only the keograms. Looking at full images will indeed bring forth a lot of new information. Consequently, it is difficult to organize into a few parameters that fit into this paper. For this reason, we consider the analysis of the full images to be out of the scope of this paper, and we will look into it in the future.

To clarify: I do not request that the Authors look into all of those questions, of course. But I think if at least one of those ideas – or another one not from this list – could be addressed, it might bring in novelty that would radically enhance the impact of the paper and make it worthy of prompt publication.

**Minor comments**

– l. 4–5 (abstract), "This emission (...) plays a big role in the study of atmospheric airglow and aurora": This statement is quite vague; it could be worth giving a concrete example of what role this emission plays.

We have rewritten this sentence to make it more accurate.

> This study conducts a statistical analysis of the aurora observed by the Swedish satellite MATS. MATS' main instrument is a telescope that performs limb imaging at six different  wavelength intervals, among them the 762 nm wavelength emission in the $O_2$ atmospheric band. This emission, even though it can not be observed from the ground,  is important at mesosphere/lower thermosphere altitudes for both atmospheric airglow and aurora. Here, some auroral

– l. 13: If possible, please provide a reference.

> The $O_2$ atmospheric band emission at 762 nm or $O_2$Atm(0-0) is produced by radiative transition of the $O_2(b^1\Sigma_g^+, v = 0)$ state down to the $O_2(X^3\Sigma_g^-, v = 0)$ fundamental state. This emission is known to be one of the main emissions in atmospheric airglow (Chamberlain, 1995) and one of the least understood auroral emissions, as we discuss below.

– l. 16: See also Kirillov & Belakhovsky (2021) for a recent work on this topic.

Thank you for sharing this recent work with us. This study focuses on the electronic kinetics of two O2 singlets, including the b1Σg+, below 100 km. It presents altitude profiles of the volume intensity of the O2 762nm emission, assuming different energies of precipitating electrons, all above 40 keV.  It also presents altitude profiles of the different mechanisms responsible for the 762 nm emission, including direct excitation by electrons, assuming energies of 1 MeV. These assumed energies are much higher than what is relevant for our study, relating to electron precipitation from the radiation belts rather than aurora. We have decided not to include the reference, since it feels like it is not relevant to what we are trying to convey in this paragraph, and could be confusing to the readers. If the reviewer thinks it should be included, we can add a sentence such as "Electron precipitation in higher energies could be dominating below the mesosphere (see recent work by \cite{Kirillov2021})." in line 16.

– l. 44–45, "and the monthly dependence of the auroral altitude": I did not find which part of the results section addresses this point.

Thank you for pointing this out. This was an outdated statement that has now been removed.

> events is presented, together with the associated Kp-index. A statistical analysis of the peak altitude in relation to the MLT and the  auroral intensity is also performed. Finally, the discussion is presented in Section 5, and the conclusions are stated at the end.

– l. 49: Generally, the term "lower thermosphere" is used instead of "low thermosphere" in the literature; please consider adopting it.

We have made the change as suggested.

> The primary goal of the Swedish MATS (Mesospheric Airglow/Aerosol Tomography and Spectroscopy) satellite mission is to determine the global distribution of gravity waves and other structures in the mesosphere and  lower thermosphere region.

– l. 77: Would it be possible to explain why a third-order polynomial was chosen for the regression in the removal of the airglow contribution? Especially in the example selected for Fig. 1, it seems that the fitted background in panel f) is very flat; it this typical? What is the reasoning behind considering a third-order polynomial in the general case?

The reason for choosing a third-order polynomial is that whenever the satellite moves towards the dayside or nightside, there is a strong variation in the background. After several tests, we concluded that a third-order polynomial worked best when handling the background in these cases. Whenever the keogram is entirely in the nightside or the dayside, the polynomial looks flat, as in Figure 1. We chose to show this event because it was a clear and clean event, which we thought would make things easy to understand. We have added a clarification in the text.

> particular case, the background variation is small. However, when the satellite moves into the dayside or nightside, the gradient is significant, and a third-order polynomial is needed for the fit.

– l. 85: Please indicate (here or later in the section) the value of the retained threshold, to ensure reproducibility of the results.

The values of the thresholds have been added. The detection algorithm is run on images taken with IR1, while the brightness is calculated from IR2 images. This has also been clarified. IR2 has a wider bandwidth and captures the whole spectrum of the O2atm(0,0) emission. This is the reason why the auroral intensity is calculated from IR2 images.

> to an altitude of about 95 km. Keograms for each orbit and hemisphere were generated. For each column of each keogram, the algorithm verifies that its average brightness (over all the pixels above row 150) is above  $15 \cdot 10^{13}$ ph $\cdot$ nm$^{-1} \cdot$ m$^{-2} \cdot$ sr$^{-1} \cdot$ s$^{-1}$. To ensure that the brightness above the threshold is not due to the upper edge of the airglow layer, the

> algorithm also verifies that even 10 rows above the top of the airglow, the brightness is  above $18 \cdot 10^{13}$ ph $\cdot$ nm$^{-1} \cdot$ m$^{-2} \cdot$ sr$^{-1} \cdot$ s$^{-1}$. If these two conditions are met, the brightest pixel of the column is selected, and it is considered a candidate sighting of aurora. The threshold values were decided after carefully inspecting several keograms with and without aurora to achieve a balance between the detection sensitivity and the frequency of spurious detection due to noise. Note that the detection algorithm is run on images taken with the filter IR1.

> For each auroral event, the brightness is taken as an average of a 3x5 pixel area surrounding the peak brightness pixel. This is done in the images taken with IR2, with the background subtracted as outlined above. It should be noted that the parameters

– Figure 1: Please add axis labels in panels e) and f), as well as a colour bar for the data shown in panels a–e). Please define also 'TPlat' explicitly (for instance in the caption). You may also consider adding in panel e) a y-axis indicating the altitude of the tangent point corresponding to the pixel row numbers.

Thank you for your comment. We have added the colorbar, axis labels, a y-axes indicating the average altitude of the tangent point and the definitions of both TPLat and TPAlt in the caption.

**Figure 1.** a)-d) Processed images taken by the limb instrument at the times indicated on top of each panel. The colormap limits are the same as in panel e). e) Keogram created by adding the central column of each image in chronological order. X-axes indicate time and latitude of the tangent point (TPLat). Y-axes represent the row number and the average (among all times) altitude of the tangent point corresponding to each pixel row (TPAlt). Vertical red lines indicate the columns corresponding to the images in panels a)-d). Horizontal lines correspond to the rows plotted in the next panel. f) Pixel values of three rows of the keogram (dots) together with the fitted background (solid line). g) 2D sketch of MATS perspective of aurora (not to scale), corresponding to the peak of the parabola in the keogram (ii.), and when observed below the airglow before the peak (i.), and after (iiiii.). The grey area represents the instrument's field of view, while the white line is the tangent line.

[Figure]

– Fig. 1 caption: One of the '(ii.)'s should be '(iii.)'.

**Right.** v before the peak (i.), and after (iiiii.).

– l. 101: Would it be possible to comment on the fact that 378 events were retained for the study, while a back-of-the-envelope calculation suggests that, during the ~81 days of MATS data used in this study, there have been approximately 4860 auroral oval crossings? Is it so that O2 762 nm auroral emission is not always present in the auroral oval? Are there limitations in the instrument's operations related to e.g. lighting conditions in the atmosphere? Are the auroral signatures very often too complex for the events to be retained by the algorithm? It would be interesting to see the temporal distribution (i.e. as a function of the date in early 2023) of the obtained auroral events, for each hemisphere.

Thank you for your question. We always observe the airglow layer in our images. To detect auroral emissions, they must be spatially distinct enough from the airglow layer, and their brightness must exceed a certain threshold. In addition, we don't consider auroral events that are too complex, i.e., auroral traces in the keogram overlapping each other. We have clarified this in the manuscript's discussion, in the paragraph on the limitations concerning this work.

> the actual auroral peak and had to be corrected manually. An estimation has revealed that in about 90% of the crossings,
> 155 the aurora is not visible or too weak to be detected by the algorithm. Then, about 2% of the crossings detect complex auroral
> distributions (indicating closely spaced multiple auroral structures) that are omitted, while about 8% detect aurora clear enough
> to keep in the study. It is also relevant to mention that due to operational reasons, some data gaps reduce the dataset to about
> 85% to 90%. Some bias might also have been introduced when removing the background airglow from the auroral events

– l. 113: How representative are the statistics in the cases where only three events are in a given data bin? Would it remove many data points if selecting a higher threshold for calculating the mean and standard error of the mean?

We have evaluated the number of data points remaining if we consider bins with a larger number of elements. With at least 3, we get 33 data points. Increasing the threshold to 4, we get 26 data points, and only 1 data point with high kp remains. Increasing it to 5, we get 24 data points, with no data points in the range 6-9 kp. Events with high kp are not common, and we wanted to keep some in the plot; that is why we chose to keep the bins with at least 3 data points and not increase the threshold.

– l. 117–118, "A notable feature of this plot is the lower altitude of events between 5 and 7 MLT, especially in the southern hemisphere": It seems to me that it is in fact *only* the case in the southern hemisphere, as the few northern-hemisphere data points are at altitudes very closed to the average values. Please correct the statement.

> events in the southern hemisphere can be seen during the night hours, while more northern hemisphere data points are visible in
> the morning and evening hours. A notable feature of this plot is the lower altitude of events between 5 and 7 MLT , especially
> in the southern hemisphere.

– Figure 2: The chosen colour map is not adequate, as it is not accessible to people with colour vision deficiencies. Please refer to the ANGEO guidelines to revise the figure (https://www.annales-geophysicae.net/submission.html#figurestables), and consider using a suggested tool such as Coblis.

Thank you for your comment. We have changed the colormap to 'inferno', which is color-blind friendly.

– Figure 3: Would it be possible to provide a measure of the uncertainty on the average altitude values in panel a)? Besides, panel b) is missing its y-axis label (name of the plotted parameter).

We have added the standard deviation/sqrt(n) as the uncertainty of the mean in the legend.

[Figure]

**Figure 3.** a)  Average altitude of  events occurring within the same one-hour bin as a function of MLT. Asterisks represent the northern hemisphere, while triangles represent the southern hemisphere. Blue includes events with Kp-index between 0 and 3, green between 3+ and 6, and red between 6+ and 9. Error bars indicate the standard error of the mean. Horizontal dashed lines indicate the averaged altitude of all events in each of the Kp-index bins. The mean values are indicated in the legend together with the uncertainty of the mean. b) Intensity as a function of altitude for all events analyzed. Colors and markers follow the same pattern as in a).

We have also added the y-axis label and changed the units to kR.

– Fig. 3 caption: "One-hour average" suggests a temporal average (e.g. from a time series), but here I think you are referring to the average of events occurring within the same 1-hour MLT bin. Please consider rephrasing to avoid ambiguity. In addition, please indicate whether the boundaries of the Kp bins are included or excluded (i.e., is the first bin from Kp = 0 to Kp = 3– or to Kp = 3? If the latter, then I presume that the second bin starts at Kp = 3+).

**Figure 3.** a)  Average altitude of  events occurring within the same one-hour bin as a function of MLT. Asterisks represent the northern hemisphere, while triangles represent the southern hemisphere. Blue includes events with Kp-index between 0 and 3, green between 3+ and 6, and red between 6+ and 9. Error bars indicate the standard error of the mean. Horizontal dashed lines indicate the averaged altitude of all events in each of the Kp-index bins. The mean values are indicated in the legend together with the uncertainty of the mean. b) Intensity as a function of altitude for all events analyzed. Colors and markers follow the same pattern as in a).

– l. 124: If using the phrase "clear correlation", please calculate a relevant correlation coefficient as part of the data analysis.

Thank you for the observation, we substituted the word "correlation" for "dependence".

This work showed a clear  dependence between latitude and Kp-index (see Figure 2). Events associated with lower Kp-index were found to be closer to the poles, while events associated with higher Kp-index were located at lower latitudes.

– l. 127–128: The statement about the auroral intensities expressed in kR is difficult to verify by looking at the figures. Would it make sense to present the spectral intensities shown in Fig. 3b directly as limb brightness using the conversion described in l. 121–122, for instance?

We have changed the units in all the plots that showed spectral intensity to auroral intensity in kR. We have removed the statement in lines 121-122, and added a sentence in the analysis instead.

> Figure 3b shows a scatterplot of the peak intensity vs the peak altitude for all events analyzed, with different colors and markers for different bins of Kp-index and hemispheres, respectively. Most events have altitudes between 95 and 110 km, and the intensities range between  200 and 5500 kR.

> **3  Analysis**
>
> 65  The data set used in this study spans from February 8 to April 30, 2023, corresponding to the period when the satellite was in continuous operation with nominal pointing. Data with tangent point latitude above 45 ° in both hemispheres are used in the study.
>
> Figures 1a-d show an example of four limb images taken by the limb instrument. The spectral intensity has been converted to auroral intensity (in Rayleighs) by multiplying it by $4\pi10^{-10}$ and the bandwidth of the filter in nm (8 nm for IR2). In these

– l. 131: Earlier (l. 114), it read '104 km'; please harmonise.

The average is 102.8 km, so we have changed the value to in l. 114 to 103 km. Thank you for spotting the typo.

– l. 134–135, "For MLTs of 5 to 7, the average altitude of the peak tends to be below average": As mentioned above, this seems to only be the case for the southern hemisphere; please rephrase.

We have specified the hemisphere for which this is observed.

> global changes in the auroral characteristics as such. For magnetic local times of 05:00 to 07:00, the average altitude of the peak in the southern hemisphere tends to be below average. This may be related to the precipitation of energetic electrons

– l. 135–136: More precisely than energetic electrons, in the morning sector, the role of pulsating aurora has been emphasised in producing lower auroral peak altitudes compared to the evening and midnight sectors (see e.g. Partamies et al., 2022, on the 557.7 nm and 427.8 nm aurora).

This is a very good point. We have modified this sentence.

> peak in the southern hemisphere tends to be below average. This may be related to the precipitation of energetic electrons  causing pulsating aurora, which have been found to be more prominent in the morning sector, and would produce lower peak altitudes (Partamies et al., 2022).

– l. 137: 'left panel' --> 'right panel'

Thank you for spotting the mistake. It has been fixed.

– l. 137–138: The reported general trend between peak altitude and brightness is not at all obvious from Fig. 3b. Further statistical processing and a revised figure would be necessary to make an assessment on this matter.

Statistically, there is a relation between the energy flux and the characteristic energy of auroral electrons (going back to Knight, 1973), which was the motivation behind our analysis. We believe this relationship may also be observable in the plot, but we agree that it is not very convincing. We have now rewritten this paragraph, adding the possible reason for the weak relation in our data.

> In Figure 3b, the spectral intensity was plotted as a function of altitude.  While events with low auroral intensities seem to have peak altitudes evenly spread between 95 and 110 km, events with high auroral intensities seem to decrease with increasing altitudes. The Knight relation (Knight, 1973) demonstrates that field-aligned electron beams with higher energy flux  are associated with higher characteristic electron energies. This can explain our results, since higher electron flux results in brighter aurora, and higher electron energy will lead to electrons reaching lower altitudes. Note that in this study, the latitudinal extent of the aurora is not resolved from the images, which can slightly affect the results.

– l. 138–139: The reference to Cattell et al. (2006) does not seem optimal, as the statement it is meant to support is not at all the focus of the cited paper. In fact, the statement does not necessarily hold – see for instance Fig. 5 of Tesfaw et al. (2022), where it is clear that energy flux and characteristic electron energy are not always following the same trend. Please revise this sentence.

Thank you for your comment. We have reformulated the statement and changed the reference to Kinght (1973). See the modified text in the previous caption.

– l. 142–143: Would it be possible to provide an estimate of how often events with a complex spatial distribution of the aurora may have been missed? It would be interesting to know for instance if the O2 762 nm aurora generally consists of a single arc or if multiple structures can be seen during a single oval crossing. If only single arcs have been retained for the study due to the event selection algorithm, this may induce a bias in the results, which would be worth evaluating and discussing in more detail.

Thank you for your comment. We have added to the discussion some sentences discussing this issue, as we discussed in the first reply to this comment.

> It is important to mention some limitations concerning this work. Our detection algorithm worked well when finding clear and isolated events, but may have missed some events with a more complex spatial distribution of the aurora, e.g., with multiple adjacent large-scale structures. During the manual visualization of the detected events, some unclear cases were removed from the database. In some clear cases of aurora, the highest pixel had been automatically selected close to the edges instead of the actual auroral peak and had to be corrected manually. An estimation indicates that in about 90% of the auroral oval crossings, the aurora is not visible or too weak to be detected by the algorithm. Then, about 2% of the crossings detect complex auroral distributions (indicating closely spaced multiple auroral structures) that are omitted, while about 8% show clearly detectable aurora. It is also relevant to mention that due to operational reasons, some data gaps reduce the dataset to about 85% to 90%. Some bias might also have been introduced when removing the background airglow from the auroral events happening during specific times when the satellite moved into or out of the sunlit portion of the orbit. For limb observations alone, it is

– Although it is considered a different emission line of O2 (1.27 μm), it would be worth referring to the recent study by Gao et al. (2020) using 18 years of SABER data, since their

methods are adjacent to yours, and it would prove insightful to discuss how your results compare to theirs. So little has been published on O2 auroral emissions that it would be worth mentioning the more recent literature addressing it.

Thank you for sharing this valuable reference. This study uses SABER limb imaging to study the O2 1270 nm emission. They provide measures on peak height, auroral intensities, peak volume emission rates, and they establish a relation between these quantities and the solar activity using the kp index. This study uses similar observations and methodology to ours and also studies O2 aurora, for which it should definitely be used for discussion in our article.

Since it is a different emission, it might not be useful to compare the auroral intensity, but the peak height and latitudinal distribution of the events should be comparable.

We have included it in the discussion in the following way:

> This work showed a clear  dependence between latitude and  SME index (see Figure 2) of 762 nm aurora. Events associated with  a weaker activity of the auroral electrojet (lower SME index) were found to be closer to the poles, while events associated with  a stronger activity of the auroral electrojet (higher SME index) were located at lower latitudes. This agrees with the results of the study on $O_2$ aurora at the 1.27 $\mu$m emission by Gao et al. (2020), who found that the $O_2$ auroral zone expands equatorward with the enhancement of the geomagnetic activity. The result also follows the well-known expansion of auroral ovals to lower latitudes during high geomagnetic activity (Kivelson and Russel, 1995).

>  energies. This can explain our results, since higher electron flux results in brighter aurora, and higher
> 155    electron energy will lead to electrons reaching lower altitudes. Note that in this study, the latitudinal extent of the aurora is not resolved from the images, which can slightly affect the results. Gao et al. (2020) performed a long-term study on $O_2$ aurora based on observations of the emission at 1.27$\mu$m. It was found that, for Kp Levels from 1 to 5, the auroral peak height varies between 104 and 112 km. In our study, peak heights have been found down to about 95 km. This discrepancy might be related to the different ways of handling the removal of the airglow layer.

**Cited references**

– Gao et al. (2020), https://doi.org/10.1029/2020JA028302

– Kirillov & Belakhovsky (2021), https://doi.org/10.1029/2020JD033177

– Partamies et al. (2022), https://doi.org/10.5194/angeo-40-605-2022

– Tesfaw et al. (2022), https://doi.org/10.1029/2021JA029880

---

## Author Response (AR2)

The Authors have carefully considered all my comments and provided satisfactory replies to each of them. They have made the corresponding changes in the revised manuscript, and I appreciate their efforts in incorporating as many of my suggestions as reasonably doable, given the limitations of the dataset and the constraints on manuscript length. Overall, I am satisfied with this new version.

I am therefore glad to recommend this manuscript for publication in ANGEO Communicates, pending a few very minor comments and suggestions are considered.

Thank you very much for your comments and suggestions. We feel that the manuscript has greatly improved thanks to them. Below, we give detailed replies to each minor comment.

- I. 93: In section 3, the IR1 filter is now briefly mentioned. It could be worth adding in parenthesis its central wavelength (presumably 762 nm?), as well as a brief explanation why this channel was preferred over IR2 for the detection of auroral emission.
  Regarding the choice of filter, there is no specific reason why we chose IR1 for the detection of the aurora. Both channels have reliable data, and in the beginning, we started analyzing the images taken with the IR1 channel. Later on, we realized that for the analysis of the auroral intensity, it might be better to use IR2, since IR1 might miss some part of the spectra, and we used IR2 for the auroral cases detected with IR1. We have modified the last paragraph in the discussion to reflect this. We have also added the wavelength and bandwidth of IR1. Thank you for pointing out that it was missing.
- Fig. 2 caption: I think a word is missing: "The solid \*line\* represents a fit of the data". I can also suggest adding that this is a \*linear\* fit for completeness.
   The typos in Figure 2's caption have been corrected. Thank you for noticing.
- I. 147–152: I appreciate the more careful phrasing of the description of the relation between auroral intensity and altitude (Fig. 3b), although I still find the trend difficult to see convincingly. Would it help visualising it if, for defined auroral intensity bins, you added the mean (or median) altitude together with a measure of the spread of the data (standard deviation or interquartile range, as appropriate)? I am thinking that linearly or logarithmically spaced intensity bins could be tried out, since the majority of the data points are concentrated around the lower end of the range of values. I leave it to you to decide if this is at all helpful and worth adding to the figure; if the result does not look convincing, it is also fine to keep the figure and text as they are.

Following your suggestion, we have added in Figure 3b the median of three auroral intensity bins: from 0 to 550, 550 to 1000, and above 1000 kR. The dots have also been plotted at the median intensity value within each bin. The error bars indicate the 25 percentile range around the median. We think these dots make what we wanted to describe clearer.

Figure 3. a) Average altitude of events occurring within the same one-hour bin as a function of MLT. Asterisks represent the northern hemisphere, while triangles represent the southern hemisphere. Blue includes events with Kp-index between 0 and 3, green between 3+ and 6, and red between 6+ and 9. Error bars indicate the standard error of the mean. Horizontal dashed lines indicate the averaged altitude of all events in each of the Kp-index bins. The mean values are indicated in the legend together with the uncertainty of the mean. b) Intensity as a function of altitude for all events analyzed. Colors and markers follow the same pattern as in a). The black dots with error bars represent the median and ± 25% altitude range of the data divided into three auroral intensity bins; from 0 to 550 kR, from 550+ to 1000 kR, and above 1000 kR.

Figure 3b shows a scatterplot of the peak intensity vs the peak altitude for all events analyzed, with different colors and markers for different bins of Kp-index and hemispheres, respectively. Most events have altitudes between 95 and 110 km, and the intensities range between 200 and 5500 kR. In addition, the data points have been divided into three bins according to their intensity, and the median altitude of each bin has been plotted together with the ± 25% range of the data from the median point. The dots have also been plotted at the median intensity within each bin.

In Figure 3b, the spectral intensity was plotted as a function of altitude. While events with low auroral intensities seem to have peak altitudes evenly spread between 95 and 110 km, events with high auroral intensities seem to decrease with increasing altitude, as evidenced by the median values. The Knight relation (Knight, 1973) demonstrates that field-aligned electron beams

– I. 176–178: Please consider whether this statement can be enhanced by reflecting that the detection algorithm used the IR1 channel.

We have modified this paragraph in the following way:

We used the While the initial detection of aurora was done using the IR1 filter, the intensity analysis was performed using IR2 channel data in this work to capture the entire auroral emission surrounding the 762 nm wavelength. This filter centered at 763's bandwith is 8 nm and captures the whole (0,0) band. Its bandwidth is 8 nm, which can also lead to some pollution from other wavelengths. In future analysis, combining all four IR limb imaging channels can be used to increase the accuracy of the measurements.

– I. 184: Please update the upper limit of auroral intensity (now stated to be 5500 kR in the earlier sections).

The upper limit of the auroral intensity has been corrected.

Data availability: Please add how the SME index data can be retrieved.

The data availability statement has been updated with the retrieval information on the SME index and Kp index.

 Acknowledgements: Please add an acknowledgement to SuperMAG for the use of SME. A standard statement according to their rules of the road (see

https://supermag.jhuapl.edu/info/?page=rulesoftheroad) is as follows: "We gratefully acknowledge the SuperMAG collaborators

(http://supermag.jhuapl.edu/info/?page=acknowledgement)."

The acknowledgements have also been updated for the SuperMAG, thank you for letting us know how this should be done.

---

## Author Response (AR3)

**A statistical study of the $O_2$Atm(0-0) aurora observed by the Swedish satellite MATS**

Judit Pérez-Coll Jiménez[1], Nickolay Ivchenko[1], Ceona Lindstein[1], Lukas Krasauskas[2], Jonas Hedin[2], Donal P. Murtagh[3], Linda Megner[2], Björn Linder[2], and Jörg Gumbel[2]

[revised manuscript text omitted]

While the initial detection of aurora was done using the IR1 filter, the intensity analysis was performed using IR2 to capture the entire auroral emission surrounding the 762 nm wavelength. This filter's bandwith is 8 nm and captures the whole (0,0) band, which can also lead to some pollution from other wavelengths. In future analysis, combining all four IR limb imaging channels can be used to increase the accuracy of the measurements.

**6 Conclusions**

A statistical study of some properties of $O_2$Atm(0-0) aurora as measured by the limb instrument on board the MATS satellite has been presented. The auroral events were distributed between 60 and 75 degrees latitude, with events with higher Kp-index located at lower latitudes and events with lower Kp-index closer to the poles, as expected. The average altitude of all the events was 103 km, and lower altitudes were observed for aurora in the post-midnight sector. Limb auroral brightness values between 200 and 5500 kR were observed.

*Data availability.* The MATS level 1b dataset (as used in this study) is stored in the Bolin centre database [https://bolin.su.se/data/mats-level-1b-limb-cropd-1.0]. The Kp index has been retrieved from the GFZ German Research Centre for Geosciences [https://kp.gfz.de/en/data]. The SME index has been retrieved from the SuperMAG website [https://supermag.jhuapl.edu/indices/].

[revised manuscript text omitted]